# Visual Learning Towards Soft Robot Force Control using a 3D Metamaterial with Differential Stiffness

**Fang Wan**
AncoraSpring, Inc.
1001 Xueyuan Avenue, Shenzhen, China 518055
`sophie.fwan@hotmail.com`

**Xiaobo Liu, Ning Guo, Xudong Han, Feng Tian, Chaoyang Song***
Southern University of Science and Technology
1088 Xueyuan Avenue, Shenzhen, China 518055
{11930807, 11930729, 11812519, 11910826}@mail.sustech.edu.cn, songcy@ieee.org

**Abstract:** This paper explores the feasibility of learning robot force control and interaction using soft metamaterial and machine vision. We start by investigating the differential stiffness of a hollow, cone-shaped, 3D metamaterial made from soft rubber, achieving a large stiffness ratio between the axial and radial directions that leads to an adaptive form response in omni-directions during physical interaction. Then, using image data collected from its internal deformation during various interactions, we explored two similar designs but different learning strategies to estimate force control and interactions on the end-effector of a UR10 e-series robot arm. One is to directly learn the force and torque response from raw images of the metamaterial's internal deformation. The other is to indirectly estimate the 6D force and torque using a neural network by visually tracking the 6D pose of a marker fixed inside the 3D metamaterial. Finally, we integrated the two proposed systems and achieved similar force feedback and control interactions in simple tasks such as circle following and text writing. Our results show that the learning method holds the potential to support the concept of soft robot force control, providing an intuitive interface at a low cost for robotic systems, generating comparable and capable performances against classical force and torque sensors.

**Keywords:** Visual Learning, Soft Robot, Force Control

## 1 Introduction

Force feedback and force control become mandatory to achieve a robust and versatile interaction between a robotic system and the physical environment, as well as safe and dependable operations in the presence of human. Direct enhancement in sensory feedback from force and torque sensors provides robotic systems with the capability to model and react to the various physical contacts [1, 2, 3]. Traditional sensing methods for force and torque rely heavily on rigid-body theories to measure and directly react to the contact dynamics. This paper aims to explore the concept of soft robot force control by using learning-based methods with visual information collected from a soft, 3D metamaterial as a robotic interface for human-robot interactions.

Sensory integration for rigidly-built robots is usually convenient to design and model using classical theories in robotics, which usually output interaction measurement of a specific point in time-series [4]. On the other hand, soft robots made from compliant materials usually show life-like, adaptive motions at a relatively low cost. However, they are challenging to model due to their non-linear mechanics, making it difficult to integrate advanced sensory. Instead, field sensing solutions, such as light [5], magnetism [6], and vision [7, 8], are usually adopted for soft robots for proprioceptive sensing [9], which promotes the application of learning-based solutions to overcome the modeling challenge [10].

5th Conference on Robot Learning (CoRL 2021), London, UK.

3D metamaterials are artificial composite structures in three dimensions with exotic material properties [11], aiming at effective material parameters that go beyond those of the ingredient ones, capable of generating desirable response through engineering design [12]. For example, recent research shows an optical lace design [5], where optical fiber networks are integrated with various metamaterial structures to achieve effective sensory integration for human-robot interaction. On the other hand, recent development in meta-neural-network provides a novel solution for passive object recognition by acoustics using metamaterial's unit-cells to produce deep-subwavelength phase shift as training parameters [13]. Periodic properties, such as electromagnetic, optical, acoustic, and mechanical ones, provide 3D metamaterials with the engineering potential towards distributed sensing for advanced robot control, where learning-based methods become an effective tool for end-to-end integration [14].

The contributions of this paper are listed as the following.

- The investigation of a soft, 3D metamaterial with differential stiffness for passive form adaptation in omni-directions in the radial plane;
- A direct vision-based sensory integration of the 3D metamaterial for estimating the force and torque using a modified resnet;
- The design integration of a marker inside the omni-adaptive metamaterial to facilitate efficient, accurate, and real-time 6D force and torque using vision-based pose tracking;
- System integration towards soft robot force control with comparable performance against commercial force and torque sensors with much-reduced hardware cost and complexity.

This paper is organized as follows. In section 2, we introduce a class of 3D metamaterial with omni-directional form adaptation, investigate its differential stiffness properties, and propose a sensory integration by adding a camera at the base. In section 3, we use different methods to learn and estimate the rigid-soft interactions of the 3D metamaterial and experiment with a UR10 e-series robot for some basic force control tasks. Conclusion, limitations, and future works are enclosed in the final section, which ends this paper.

## 2 A Soft, 3D Metamaterial with Differential Stiffness

### 2.1 3D Metamaterial Design with a Soft Network Structure

We adopt a network structure made from soft rubber in a hollow, cone-shaped 3D metamaterial as the design of interest, which has been researched in a few recent publications to demonstrate its omni-direction adaptation for grasping [15] and sensing [16, 17]. As shown in Fig. 1, the proposed 3D metamaterial features a gradually shrinking cross-section from the base to the tip, which can be designed with various geometric layouts based on the desired adaptation and form factor. Each layer comprises several nodes that are linked by soft, compliant rubber bars. The gradually shrinking design usually ends at the tip, which can be a single or multiple nodes.

As illustrated in Fig. 1, the proposed 3D metamaterial is capable of three major adaptive modes, including surface adaptation, twisted adaptation, and edge adaptation in all directions in the radial plane. Due to material softness, the 3D metamaterial also exhibits smooth transition between the three adaptive modes during physical contact. The adaptive effects, which apply to all three adaptive modes, are observed to be the most significant around the central area along the axial direction, but less so around the base with sizeable passive reaction force and less so around the tip with less passive reaction force. During the process of form adaptation, due to the shrinking design characteristics, the tip of 3D metamaterial generally remains (or near) the exact location with only spatial twisting, exhibit a passive form-closure effect that is similar to caging in grasping, making it a simple, convenient, and practical design for robotic manipulation with unstructured environment [15, 17]. On the other hand, the network structure with a hollow space inside provides abundant design space for sensory integration [16, 18].

### 2.2 Radial Stiffness Distribution along the Axial Direction

From a mechanics perspective, one could view the soft, 3D metamaterial as a single piece of material to study its basic displacement responses under external force. Due to overall design characteris-

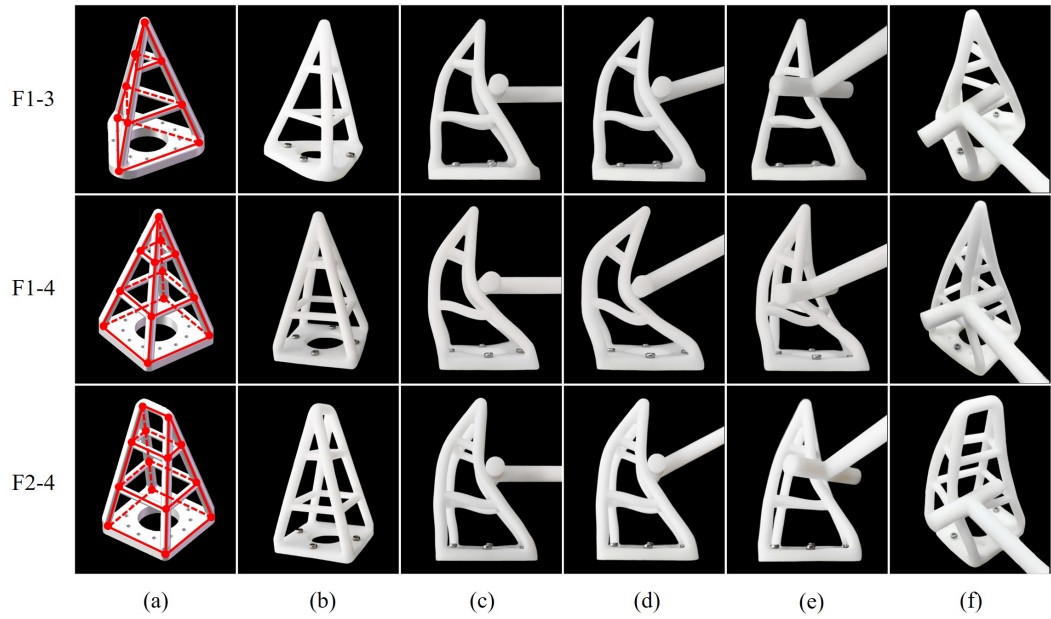

F1-3

F1-4

F2-4

(a)   (b)   (c)   (d)   (e)   (f)

Figure 1: The metamaterial design with a soft network structure: (a) node placement, (b) sample designs, (c) normal-to-axial push, (d) normal-to-surface push, (e) when pushed on the edge, and (f) twisted when pushed sideways.

tics, we conveniently viewed the proposed design as a passive robotic finger to examine its unique stiffness performance. We developed a simple test rig in Fig. 2(a) to measure the finger's radial stiffness, and reorient the finger in Fig. 2(b) to measure its axial stiffness. Three simple design variations are used as sample designs for testing, including the F1-3 design in Fig. 2(c), F1-4 design in Fig. 2(e), and F2-4 design in Fig. 2(g). We designed these three samples to share the same height of 75 $mm$, the same 40 $mm$ size at the bottom, the same number of three-layer designs in the middle, made from the same rubber material, and the same radius of the rubber bar in 2.5 $mm$. Small through-holes are also added to each finger sample for the ease of fixture on the test rig.

Based on previous observation of the finger's adaptive form response, we picked three different locations along its axial line marked in red, including the $2/5$, $3/5$, and $4/5$ location of its total height, to perform the radial stiffness test. The results are plotted in Figs. 2(d), (f), and (h). To measure radial stiffness, soft fingers were mounted on a platform vertically and a bar horizontally pushed the fingers with 1mm, 2mm, 3mm, 4mm, 5mm displacement on $2/5$, $3/5$, $4/5$ location as showed in Fig. 2(a), the corresponding forces were recorded by a VICTOR 100 digital force gauge. To measure the axial stiffness, soft fingers were mounted horizontally showed in Fig. 2(b) and a hook horizontally pulled the fingers with 1mm, 2mm, 3mm, 4mm, 5mm displacement, and the forces were also recorded by the digital force gauge.

As the result shown in Figs. 2(d), (f), and (h), the relationship between force and displacement is linear in the same locations. Then a first-order curve fitting method is used to calculate the line slope as stiffness. In some cases, the forces are null with small displacement (1mm and 2mm) as the digital force gauge is not sensitive.

## 2.3 Differential Stiffness towards Omni-directional Adaptation

A significant design feature of the proposed finger is a network structure of three-dimension interaction, which is different from existing designs that are usually limited to two-dimensional adaptation [19, 20, 21, 22] making it a soft, 3D metamaterial as a whole. As shown in Figs. 2(d), (f), and (h), results show that these finger samples exhibit consistent stiffness distribution in general, where the radial stiffness at $3/5$, or the central location, is measured to have the smallest stiffness. The radial stiffness is much increased at both $2/5$ and $4/5$ locations, which are of similar magnitude for each

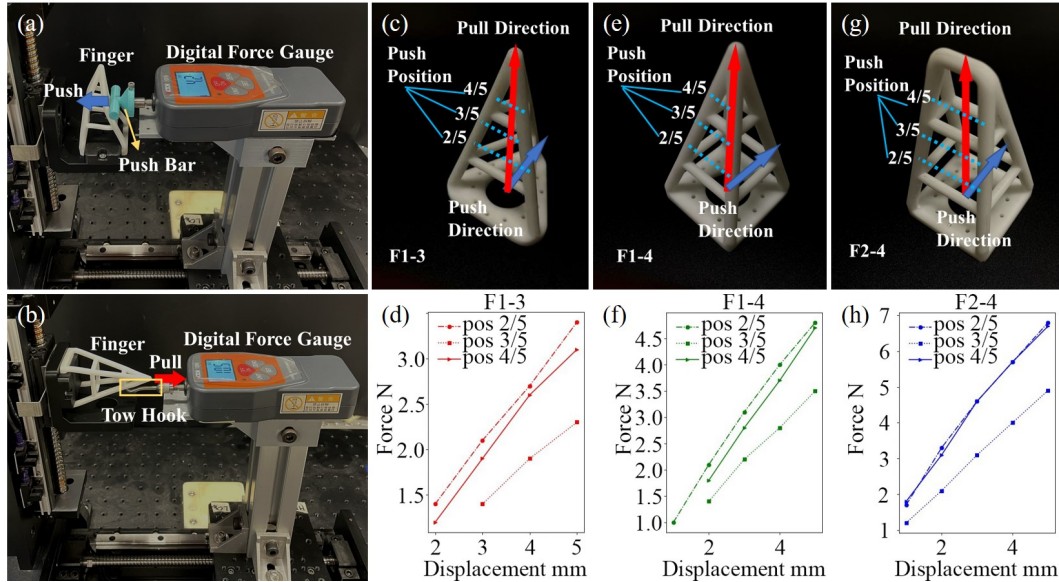

Figure 2: Differential stiffness of the metamaterial that enables form adaptation: (a) radial compression experiment setup; (b) axial stretch experiment setup; (c), (e), (g) three fingers used in the experiments: F1-3, F1-4 and F2-4. The red arrow is the pull direction to measure axial stiffness in (b), the blue arrow is the push direction to measure radial stiffness in (a), and cyan dotted lines are four loading positions which are quintiles of the finger. (d), (f), and (h) are corresponding displacement-force curves of the fingers (c), (e), and (g).

design variation. On the other hand, when more nodes are used in the design, more rubber bars will be involved in the resultant finger, and overall radial stiffness becomes much larger.

The results are shown in the left side of Fig. 3, which may be used to explain our soft finger's omnidirectional form adaptation in the radial direction during the interaction. Here, we introduce the stiffness ratio of the soft finger, measured by the ratio between the finger's axial and radial stiffness. The soft finger generally exhibits a relatively lower stiffness ratio near the central but higher readings near both ends, consistent among the three sample designs. When interacting with the environment in the radial direction, such stiffness ratio enables the finger to easily deform towards the central part while portions near both ends deform much less. Furthermore, due to its axial symmetric design, such behavior can be easily observed in any radial direction of the finger, making it omni-directional adaptive.

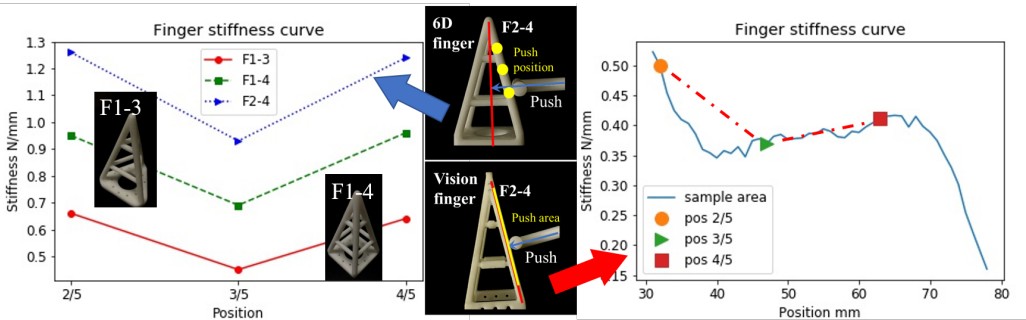

Figure 3: Stiffness distribution of the 6D finger on the left and the vision finger on the right. Left: the stiffness distribution of three vision fingers with horizontal loading. Right: the stiffness distribution of a 6D finger with surface normal loading.

While the above test only measures reading on scattered points using simple force scales, we conducted a more detailed measurement using Yaskawa's MotoMINI to conduct the same experiment automatically. In this experiment, we collected more refined measurements using an ATI's nano25

6-axis force and torque sensor measuring at the finger's base while the robot pushes along the normal direction to the finger surface along the whole length. A more slender design is tested with results plotted on the right of Fig. 3. The marked points correspond to the same 2/5, 3/5, and 4/5 locations of this particular soft finger, which exhibit similar distribution comparing to the F2-4 design on the left, but slightly skewed to the base side with higher stiffness at 2/5 location than the 4/5 location towards the tip. The results on the right side of Fig. 3 indicate that the proposed soft finger may exhibit more complex behaviors in stiffness ratio along with its geometric form, making it extremely challenging to model precisely using analytical method. In order to better leverage its omni-adaptive capability, we will introduce two different data-driven methods to integrate the soft finger with machine learning, achieving similar performance in fundamental interactions as robot force control.

## 3 Learning Soft Robot Force Control

### 3.1 Sensing Differential Stiffness with Machine Vision

Rigid-soft interaction can be hard to model, as it involves infinite degrees of freedom. To make the research feasible, we turn to the deep learning method at first sight. Deep learning constitutes a modern technique for solving computer vision problems, with remarkable results and large potential. In order to make the best use of the data-driven method, we designed a highly integrated vision-based soft finger, as is shown in Fig. 4(a). With a USB camera installed inside the structure of the soft finger, it can be easily recognized different interaction events based on merely deformation images captured, such as Fig. 4(b),(e) and (f). In this highly integrated design, we fix the soft finger with a high-precision F/T sensor ATI Nano25 [23], which is used for providing ground truth F/T data during the interaction. With this vision integrated design, we attempt to solve the rigid-soft interaction problem by using a learning-based method.

ArUco tag has been widely used in the computer vision and robotics community for its simplicity and low cost. In this paper, we also explored the possibility of learning the rigid-soft interaction of the soft finger by tracking the ArUco tag installed on the finger, as shown in Fig. 4(e). The USB camera captured gray-scale images of $640 \times 400$ pixels at 120 frames per second (fps). Figs. 4(f), (g), and (h) show the camera captures of the ArUco code when the finger is without external force, with pushing on one side and with twisting around its vertical axis, respectively. The USB camera was connected to a laptop with eight Intel i7 processor running at 2.8GHz. With the ArUco tag placed at 26 mm away from the camera, the detection function in OpenCV [24] was able to achieve a successful detection rate of 100% and the computational time cost is 3 ms per image on average. By examine 15,000 images of the at the same pose, we found that the perturbation range of pose estimation is $[0.18, 0.06, 0.0008]$ mm for $x, y, z$ and $[0.086, 0.131, 0.004]$ degrees for roll, pitch, yaw. In the rest of the paper, we always applied a moving average to the detected pose within a window length of 5.

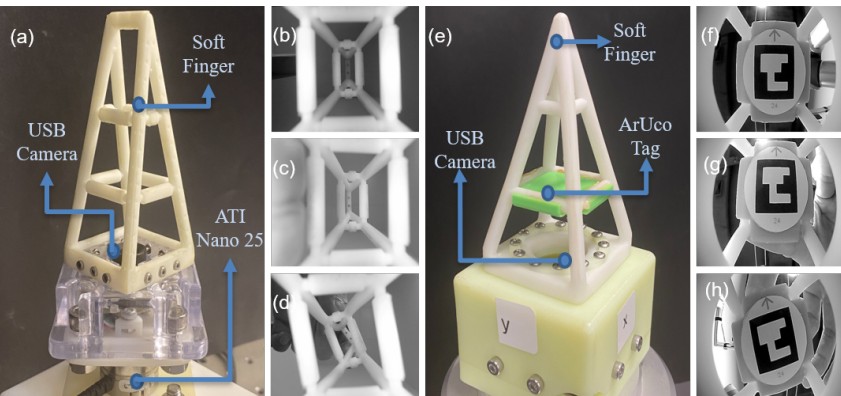

Figure 4: Vision integration with a camera mounted on the base: (a) Vision integration A; (b) default camera view of A; (c) camera view of A during normal interaction; (d) camera view of A during twisting interaction; (e) Vision integration B; (f) default camera view of B; (g) camera view of B during normal interaction; (h) camera view of B during twisting interaction

## 3.2  Learning Rigid-Soft Interaction

During the interaction with the physical world, the soft finger will deform due to the contact force/torque. The idea of learning a neural network that maps sensor readings to 3D force estimation through labeled training data has been adopted in previous research [1]. The problem in this paper is to estimate the force and torque $ft = (Fx, Fy, Fz, Mx, My, Mz) \in \mathbb{R}^6$ at the base of the soft sensor from the deformation of the metamaterial. Due to the complicated property of the metamaterial, we use an image of the internal structure $I$ to represent the deformation and use a convolutional neural network to learn the mapping $ft = F(I)$.

In order to automatically collect the contact information with the minimal human invention, we build an automated data collection platform, as is shown in Fig. 5(a). To obtain high-quality controllable contact data, we choose Yaskawa Motomini [25], an industrial desktop robot arm capable of achieving $0.02\ mm$ repeatability. We 3D print a T-Shape indenter and install it on the flange of the robot arm. As for the contact data measurement, we choose ATI Nano25, an expert industrial 6-axis F/T sensor, to record the contact force/torque at the soft finger base during the interaction. A commercialized USB camera captures the deformation image with a resolution $640 \times 480$, which is easily installed inside the soft finger.

We use ROS [26] Yaskawa driver to control the movement of the robot arm. The indenter is commanded to make indentation perpendicular to the surface and move along the surface axis from bottom to tip of the soft finger. The waypoints along the surface axis are 1mm spaced, and the robot will move along the surface's normal direction (pointed into the surface) at a depth of 1mm, 2mm, and 3mm, at every waypoint. When the robot has moved to the target position, the camera and the F/T sensor will record a pair of data points. We repeat the indentation test 5 times for every surface. As a result, we collected more than 2,500 data points and trained a convolutional neural network.

Convolutional Neural Network has become an effective method to deal with computer vision problems. The CNN structure is shown in Fig. 5(b). We deploy the residual blocks [27] as the feature extraction layers in the hope of capturing the deformation features of the soft finger during the interaction. Then two fully connected layers are connected to the feature extraction layers, followed by a sigmoid activation function to get normalized predict results between [0, 1]. As we use the same feature extraction layers to make regression on multi-variables, it is helpful to normalize the data before feeding it into the neural network. We use the CNN to directly predict the force along the $x$-axis and $z$-axis and torque along $y$-axis due to the contact condition. We use Adam Optimizer and train the network with mean squared error as a loss function.

We hold 10% collection data as the test set. The comparison result between visual force CNN prediction and ground truth is shown in Fig. 5(c), (d), and (e). The coefficient of determination $R^2$ of the prediction along $x$-axis, $z$-axis, and torque along $y$-axis are listed in the figure. The figure shows that our visual force CNN is capable of making precise predictions, with $R^2 > 0.9$ in each dimension. However, our model also shows its deficiency when making contact with objects other than the T-Shape indenter. The lack of generalization ability leads to bad deployment results when directly using its prediction as an F/T sensor. Therefore, we would like to explore a more robust way to infer contact information based on internal vision sensing.

## 3.3  Tracking Interaction using 6D Pose

The deformations of the soft finger by external forces lead to the displacement of the tag from its original pose represented by $(\delta x, \delta y, \delta z, \delta roll, \delta pitch, \delta yaw)$. Thus the soft finger might be a good joystick to teach the robot. When installed on the tool flange, the soft finger could achieve results similar to admittance control with a force/torque sensor. Considering that the soft finger is hardly affected by force along the $z$-axis, we designed a positional control strategy. We use the SPEEDL API from UR and control either the position of tool center point (TCP) or its orientation. In the former mode, the linear speed $v_x$ and $v_y$ is proportional to $\delta x, \delta y$ while $v_z$ is proportional to $\delta yaw$. Similarly, in the latter mode, the rotational speed is proportional to $\delta x, \delta y$ while $v_z$. The 6D displacement of the tag is transformed to the frame of the robot base from that of the camera. Hence the interaction is very intuitive to the human user.

We conducted control experiments using a UR10e robot arm, from which we can read the force/torque at the TCP directly from the controller. The task is to drag the TCP so that it follows a particular path on a horizontal plane as shown in Fig. 6(a)-(d), namely a printed circle and the

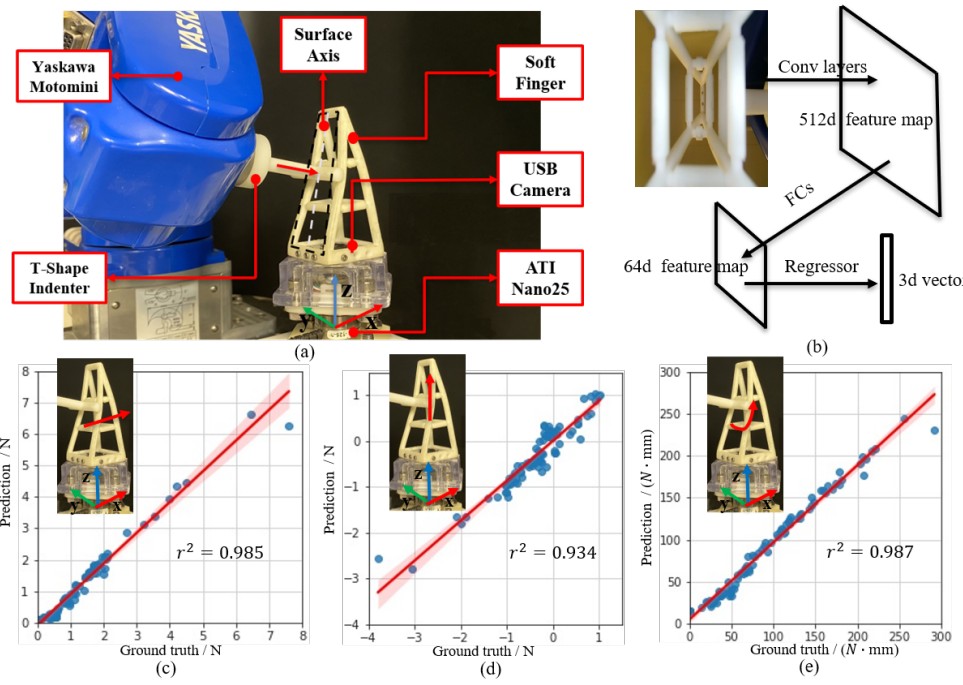

Figure 5: Learning rigid-soft interaction on the surface normal: (a) experiment setup; (b) network design; (c) Force prediction result along x-axis on test set; (d) Force prediction result along z-axis on test set; (e) Torque prediction result along y-axis on test set.

abbreviation of Conference on Robot Learning. We record the positions of the TCP while we taught the robot and plotted the resulting paths using a force-torque sensor and the soft finger. The soft finger has demonstrated excellent experience of human-robot interaction and was able to complete the tasks well.

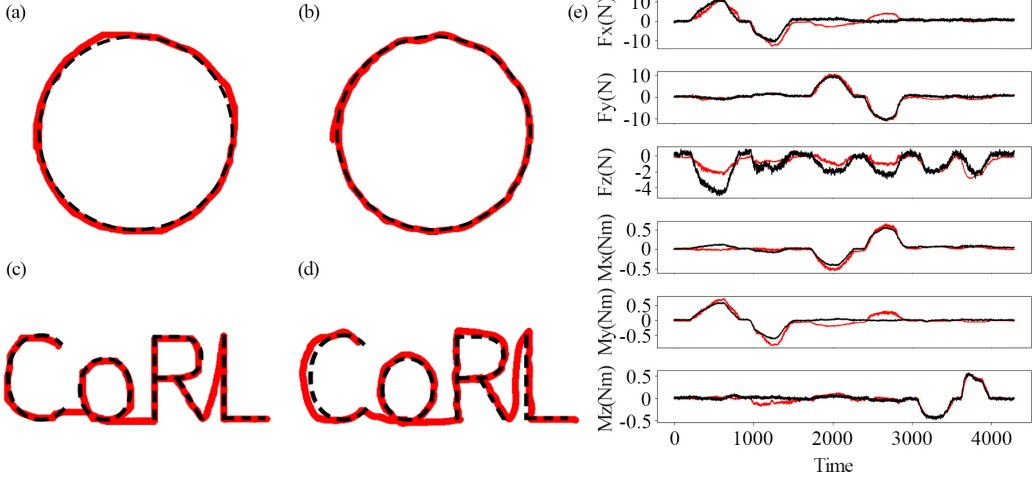

Figure 6: Tracking interactions by visual estimation of the internal 6D pose: (a)(c) paths completed using the soft finger; (b)(d) paths completed by teaching the robot using force/torque sensor on UR10e; (e) real-time force and torque predicted by the learned model in red against those obtained from the UR10e's controller in black.

A further hypothesis is that the 6D displacement of the tag could well represent the deformation of the soft finger under external force. Hence a mapping from such displacement to the force could be learned using the machine learning method. We installed a soft finger on the tool flange of the

UR10e. We wrote a program to automatically record the images captured by the USB camera and the force/torque values read from the robot controller while we push and twist the soft fingers by hand. With the high fps camera, we collected a total of 30000 images and their corresponding force/torque values in a few minutes. We used a single regression model of three hidden layers with 200, 100, and 6 neurons, respectively, and the input is the 6D displacement of the tag, and the output is the 6D force and torque. The loss function is mean squared loss, and the optimizer is Adam with a learning rate of 0.001. The regression converged quickly after a few epochs.

To test whether the learned model can be transferred to a new sensor of the same design, we have fabricated two fingers of the same design, material, and fabrication methods. One was used to collect the training data as described above and was later replaced by the other finger to test the performance. The standard deviations of the predicted signals $(Fx, Fy, Fz, Mx, My, Mz)$ of the soft sensor measured over 100 seconds are $(0.009, 0.034, 0.007)N$ and $(0.0013, 0.0009, 0.0010)N \cdot m$. The same measurement for UR10e are $(0.323, 0.431, 0.253)N$ and $(0.0123, 0.0107, 0.0058)N \cdot m$. The noise level of the soft sensor is around 10 times lower than that of the UR10e's. Fig. 6(e) shows the real-time force and torque predicted by the learned model in red, and the values obtained from the robot controller in black. The model can give a fairly good prediction even with many discrepancies between the prediction and the read values from the controller.

## 4 Final Remarks

In this paper, we explored the visual integration of a soft, 3D metamaterial with an omni-directional adaption by adding a monocular camera at the design base to adopt a data-driven method for human-robot interaction. We systematically investigated the metamaterial's design characteristics, which feature a radial stiffness ratio with differential distribution along its axial direction, resulting in an omni-directional adaptation that is challenging to model, yet attractive to interact with. We proposed two different design enhancements to the original finger design by simply adding a camera at the base or fix another fiducial marker inside. We built several learning models to investigate the mapping between the finger mechanics from vision and the force and torque response from the interaction. Our results show that it could become a promising direction to use data-driven learning method to leverage the modeling challenge of soft robots for advanced robot control and interaction, where a potential future for soft robot force control may be of interest to researchers seeking more adaptive, versatile, and low-cost sensing and interaction of their robotic system.

As the soft sensor senses through visual signals, its output frequency is determined by the frequency of the camera and the inference frequency of the prediction model. In our experiment, the frequency is currently limited by the frequency of the camera, which is 120 Hz and is already higher than most previous research tactile sensors employing camera [28]. This is comparable to the commercial product FT 300 from Robotiq, which is 100 Hz. As the computation time of ArUco tag detection is 3 ms on average, the upper bound could further increased to 333 Hz with higher speed camera. In general, tactile sensor employing camera suffers from the relatively low frame rate and higher computational cost compared with other conventional sensor. However, it made the proposed soft sensor to be more accessible and easily assembled with much lower cost. The total cost of such a soft finger with sensing capability is lower than 100 USD.

As reviewed by [28], most of the previous research in tactile sensors employing camera converted the physical contact into light signal based on some kind of micro or local deformation of the contacting point and the camera was placed within an controlled environment with stable lighting devices such as LED. One of the major difference of our proposed soft sensor is that the physical contact to light conversion is based on large deformation of the soft metamaterial. In order to investigate the soft metamaterial alone, we kept it bare and did the experiment in daily lighting condition in the lab without any extra lighting devices. The stability could be harmed if the lighting condition become weak, but can be engineeringly improved with low-cost artificial lighting when necessary.

Due to the limited precision of the force/torque sensor build in UR10 e-series, it's not worthwhile to compare the model prediction to the values read from controller. We plan to collect more data and use force/torque sensors with higher precision to better characterise the soft fingers during interaction. Looking for a potential low-cost replacement of the industrial force/torque sensor, we will further improve the design of the soft sensors to overcome some of the current limitations and explore promising application scenarios.

**Acknowledgments**

The authors would like to acknowledge the constructive comments provided by the reviewers and editors. This work was jointly supported by National Science Foundation of China (No. 51905252), Shenzhen Institute of Artificial Intelligence and Robotics for Society Open Project (No. AC01202005003), Shenzhen Long-term Support for Higher Education, and AncoraSpring, Inc.

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
