# OpenReview forum: "Visual Learning Towards Soft Robot Force Control using a 3D Metamaterial with Differential Stiffness"
_robot-learning.org/CoRL/2021/Conference — CoRL2021 Poster_

### Official Review · Reviewer_mjxa · 2021-07-19

**Originality:** Good
**Technical Quality:** Good
**Clarity Of Presentation:** Very Good
**Impact:** 4

**Recommendation:**

Weak Accept: I recommend accepting the paper, but will not argue for my recommendation if the majority of other reviewers have a different opinion.

**Summary:**

This paper explores the feasibility of learning robot force control and interaction using soft metamaterial and machine vision.

The paper explores two methods: directly learning the force and torque response from raw images of the metamaterial’s internal deformation, and indirectly estimating the 6D force and torque using a neural network by visually tracking the 6D pose of a marker fixed inside the 3D metamaterial. The paper finally integrates the two methods and achieves similar force feedback and control interactions in simple tasks.

The results show that the learning method holds the potential to support the concept of soft robot force control, providing an intuitive interface at a low cost for robotic systems, generating comparable and capable performances against classical force and torque sensors.

**Issues:**

No issue.

**Reviewer Expertise:**

Poor: Limited knowledge of the area

**Strengths And Weaknesses:**

I have no knowledge in this domain before. I cannot evaluate the novelty and technical contribution of the paper correctly. I only make sure the claims in the paper are correct.

Strength:

The paper has several contributions:
-- The investigation of a soft, 3D metamaterial with differential stiffness for passive form adaptation in omni-directions in the radial plane;
-- A direct vision-based sensory integration of the 3D metamaterial for estimating the force and torque using a modified resnet;
-- The design integration of a marker inside the omni-adaptive metamaterial to facilitate efficient, accurate, and real-time 6D force and torque using vision-based pose tracking;
-- System integration towards soft robot force control with comparable performance against commercial force and torque sensors with much-reduced hardware cost and complexity.

All the contributions are clearly stated and empirically demonstrated in the paper.

Weakness:

I do not see explicit weakness in the paper.

**Summary Of Recommendation:**

I cannot evaluate the paper correctly so I give a rough evaluation of the paper. I will discuss with other reviewer about the final recommendation for this paper.

---

> ### Author Response · Authors · 2021-08-27
> **Reply to Reviewer mjxa**
>
> We sincerely appreciate the reviewer's positive comments on our work and modesty in the review. The reviewer made an exact observation of the contributions made in our paper. The overall goal of this paper is to introduce a potential research direction in soft robot force control by integrating a learning-based method and metamaterial design for human-robot interaction, which could be further expanded to robot learning in manipulation but not yet explored with much literature. We have made extensive revisions based on comments from the other reviewers and AC. We believe the revisions made significantly improve the quality of our paper, and we hope the reviewer would appreciate our revised manuscript as well.

---

> > ### Comment · Reviewer_mjxa · 2021-09-03
> > **Response**
> >
> > Thanks the authors for addressing my concerns. I keep my score.

---

### Official Review · Reviewer_avKF · 2021-07-19

**Originality:** Good
**Technical Quality:** Good
**Clarity Of Presentation:** Good
**Impact:** 3

**Recommendation:**

Weak Accept: I recommend accepting the paper, but will not argue for my recommendation if the majority of other reviewers have a different opinion.

**Summary:**

The paper applies a deep learning approach utilizing convolutional neural networks to the problem of extracting torque measurements from a soft structure. This structure, having a pyramid-like form, is meant to be attached to the end-effector of a robot. Images and videos show an example where it is mounted on a UR 10 robot arm. The general idea is to record camera images of the internal deformation of the structure from its base and then using a supervised learning approach to learn a neural network predicting the external force. The motivation for this is the idea that the structure can be used as a 'soft finger' on a robot to gently guide it for demonstrating/recording movement trajectories.

**Issues:**

- Figure 5: The graphs should show a grid-pattern to allow the viewer to evaluate the deviation between predicted and actual force. Furthermore, both x- and y-axis should have the same range.
- Section 3.2 should be restructured and introduce the exact learning problem in a formal way.
- Explain what the exact problem the paper is trying to solve is. Why do we need deep learning for it? How can it be made applicable to the outside of your lab and to other researchers who might want to use the proposed finger structure but do not have access to the testbed used to collect the supervised data?
- Is the proposed approach using CNN networks really the only approach capable of solving this problem? For example, how does the proposed algorithm using a camera compares against directly attaching a force/torque sensor to the base of the finger structure?
- Fig. 6 should show in (a-d) the reference trajectory to follow.
- Lastly, why use a supervised learning approach + a controller in task-space? Why not directly learn the motor commands given the deformation in the finger structure?
- The paper could discuss further camera-based approaches for manipulation such as "OmniTact: A Multi-Directional High-Resolution Touch Sensor"
- The paper would also benefit from a better discussion about the potential applications of this structure, especially regarding the prospect of using it for human demonstrations in the robot learning community.

**Reviewer Expertise:**

Good: General knowledge of the area

**Strengths And Weaknesses:**

Although not entirely novel (see below), the idea of extracting force measurements from visual observations is a good line of enquiry for this paper. Furthermore, the general setup of the experimental testbed seems rigorous. The video is of very good quality.

However, I have major concerns regarding the general learning setup and its motivation as it is described in the paper.
- The general learning problem is not very well formalized: What exactly is the learning problem the paper tries to tackle? Just as an example, the paper does not mention that the learning setup is supervised at least once. The greater question is, why is the learning problem supervised? Why not unsupervised or using reinforcement learning? Furthermore, the ATI Nano25 sensor appears smaller than the camera box at the bottom of the finger. Why not simply attach the ATINano25 sensor instead of a camera? What extra information would the camera provide, and why would we be unable to use an F/T sensor at the base?
- The estimated force/torque data is utilized to be fed back into the controller of the UR10 in task-space, controlling the position of its end-effector. Why can this not be solved with said additional force sensor at the base of the finger or a more sensitive (robot) controller?  This setup would not require deep learning at all.
- Why is deep learning the algorithm of choice? Why not use the information from the tag extracted with OpenCV to train an SVM or another regression algorithm? Figure 5 shows only the predictions of the used neural network, some other learning algorithm as baseline would be expected. Or an explanation of why other algorithms are not applicable.
- I cannot entirely agree with the conclusion that the proposed setup would allow other researchers a more versatile or low-cost setup: Assuming reasonable variance in manufacturing or 3D-printing the proposed structure, every lab would have to carry out the calibration method described, i.e. collecting a supervised data set with force measurements and images showing the deformation. This process would also be required to be repeated every time a new finger/structure is produced.
- The paper introduced the soft structure as a 3D metamaterial. As the introduction of the paper already mentions, a meta-material is usually a composite material, i.e. a material made up of at least two or more materials. It is not clear if this is the case here, or if simply the structure of the elastic material (in the form of a pyramide like tubing) determines alone its properties.
- For me, the results presented in Figure 6 (b) (UR 10 force sensor) look considerably more natural and better than (d) (proposed camera-based approach).

A more interesting approach would have been to learn the required force/torque estimated in a reinforcement learning setup, in the absence of any additional sensory equipment, by letting the robot arm interact in some fashion with the environment. This would also strengthen the applicability of the paper for a robot learning conference.

**Summary Of Recommendation:**

From the point of robot learning, it is not clear what the exact contribution of the paper is. The described learning setup is rather weak, and the principal problem could be solved by exchanging the camera with a force/torque sensor and a more sensitive robot control algorithm, which would not require learning at all. Furthermore, the approach is limited to setups where it is possible to acquire a supervised data set by the use of force/torque sensors. A more exciting approach could be an autonomous learning process in which the robot arm learns the direct mapping from captured structural deformation to motor commands. This would also allow the operator to apply the proposed finger structure to any robot, not just ones with accurate kinematic modelling in the absence of any force/torque sensors.

---

> ### Author Response · Authors · 2021-08-26
> **Reply to Reviewer avKF (Part IV)**
>
> For the issues raised by the reviewer, please refer to the following. We sincerely thank the reviewer's constructive comments on our paper, which are all addressed and revised as suggested.
> 1. We have revised Fig. 5 as suggested by the reviewer, which is a nice improvement to our paper.
> 2. We have restructured the learning problem in section 3.2 as suggested by the reviewer, which clarifies our work formally.
> 3. We have clarified the reason for using the learning method in our design, as shown in the updated section 3.2, paragraph 1. Please refer to our detailed reply to the third point raised by the reviewer.
> 4. As we explained above, CNN is not the only solution but could serve as an obvious one as a baseline for the learning-based method in a general sense. We have patented our design and plan to make it freely available to the research community soon, once we have standardized the process of calibration, which is an ongoing work to be published soon.
> 5. We have added the reference trajectory, as recommended by the reviewer.
> 6. We agree with the reviewer that direct learning the motor commands could be an exciting subject to explore, which is one of the following works we are trying to resolve. For the purpose of this paper, we aim to provide a low-cost soft solution providing 6D force and torque, which could be a direct replacement for the traditional force/torque sensor in existing applications. As this proposed method and design is still new to the research community, we would like to share our progress through this paper with a more relevant research community. We are glad to see the various constructive comments and suggestions from the reviewer, other reviewers, and AC. We hope to publish our preliminary work through CoRL and keep working with the community on this interesting subject in our future work.
> 7. We have added a citation to the recommended paper, which is a good reference to our research. However, our current work focuses more on the 6D interaction force and torque, which is not the same as tactile or touch sensing in localized regions.
> 8. We have revised the discussion section to discuss the potential application of our proposed design in a human demonstration, as recommended by the reviewer.

---

> > ### Comment · Reviewer_avKF · 2021-08-28
> > **Reply to author's response**
> >
> > I appreciate and thank the authors for their detailed response ahead of the end of the discussion period.
> > Based on the response I have the following follow-up comments the authors may wish to address before the discussion period ends on Monday:
> >
> > >[Reply 2] As mentioned in the reply above, one can surely use an FT sensor at the base, only more expensive, less adaptive to the changing environment, and more complex in engineering design and integration. However, what if we want to use the gripper under the water, just for the sake of argument, do we need to change/waterproof the entire design. This is only a simple example, but with our design, the problem could be solved more quickly without changing the design, which is an added benefit of our method that could work both on land and under the water or in other challenging situations.
> >
> > This is not a really convincing scenario. I do no completely follow why it would be easier to waterproof a camera than a force sensor. Would this not make the task of reliably measuring deformation harder in a camera-based setup due to the change in vision underwater?
> >
> >
> > > [Reply 5] We would like to clarify that meta-material is different from composite material, as cited in reference [11] in our paper. Meta-material leverages the structural design and geometric feature, which usually involves just one material (but not limited to), to achieve mechanic properties not capable by the original material. Composite ones, as explained by the reviewer, involve at least two or more materials. Our design shown in this paper is a meta-material that involves only one material, but we are also experimenting with other designs that involve multiple materials to build such meta-structure.
> >
> > I would like to clarify that when initially reading the paper, it was not clear to me if the material the finger is manufactured from is a single type of material or a composite material. The introduction of the paper states that
> >
> > > 3D metamaterials are artificial composite structures in three dimensions with exotic material properties [11], aiming at effective material parameters that go beyond those of the ingredient ones, capable of generating desirable response through engineering design [12] [.],
> >
> > which mentions explicitly composite structures, which implies laminates and potential combinations of multiple materials.
> > It would benefit the reader if it were clearly stated which type of material is used. Technically, the authors are correct to use the term metamaterial in their paper if they use a single type of material and vary its structure. However, since the paper does not vary the structure of the material (ie on a micro or nano scale) it would be clearer to speak of an optimization of structure instead of a metamaterial optimization. An example of what I mean is given in this paper [1], which presents fine-granular metamaterials used in soft robotics. The paper under review varies the structure of the device on a rather large scale, without repetition or use of cells, while the title and abstract seem to imply the optimization of the structure of the material itself used to construct the finger.
> > Hence, it would be clearer if you speak of optimizing the structure/architecture of the device, instead of saying you optimize a 3D metamaterial since the type of material you use does not change, neither does its internal structure (i.e. the radius of the hollow segments, the material type or structure between the individual segments).
> > You could and probably should address this in section 2.1 where you introduce the architecture of the device, stating which exact type of material you use, clarify the scale on which you change the architecture (cm, based on line 92) and clarify in the abstract that you vary the structure of a 7.5 cm high robot finger. Other papers in the area of metamaterials used in soft robotics utilize commonly much smaller and repetitive structures, composed of individual cells or macro/micro structures.
> >
> > The paper's aim and the problem it is trying to solve still seem interesting, but using the term metamaterial in the title, abstract, and introduction appears to oversell the problem the authors considered in their actual experiments.
> >
> > [1] Khajehtourian, Romik, and Dennis M. Kochmann. "Soft Adaptive Mechanical Metamaterials." Frontiers in Robotics and AI 8 (2021): 121.

---

> > > ### Author Response · Authors · 2021-08-28
> > > **Reply to Reviewer avKF on Follow-up Comments**
> > >
> > > We sincerely thank the reviewer's prompt reply and comments.
> > >
> > > > This is not a really convincing scenario. I do no completely follow why it would be easier to waterproof a camera than a force sensor. Would this not make the task of reliably measuring deformation harder in a camera-based setup due to the change in vision underwater?
> > >
> > > In the underwater scenario example, from a mechanical perspective, waterproofing a camera will be much easier than a force sensor in many ways. The cost will be much lower when waterproofing a camera, a relatively mature technology common in recreational or professional photography underwater. For example, products such as GoPro include a waterproofed case that one can easily buy online. For the small camera used in our case, it is simple to custom design and fabricate a case solid enough to withstand pressure underwater within an extra 100 dollars. But a waterproofed force and torque sensor usually require specialized design integration from selected vendors that are not easily accessible. We have an underwater version of ATI's Nano17 and another common version of ATI's nano25 in our lab. Each cost nearly 10K USD (the underwater version is much more expensive) with specialized controllers (extra cost) in bulky sizes. You can customize a smaller controller for these ATI sensors (but tricky), but waterproofing these electronics would be another difficulty.
> > >
> > > As you dive deeper under the water, the static pressure increases significantly, which would interfere common force-torque sensor's performance (as there is usually a sealed chamber inside that would cause noise signals in sensor reading if the environmental pressure changes). However, if using our proposed design, the resultant performance is mainly related to the lighting condition (can be easily integrated underwater), camera specs (protected by the solid case), soft structure's mechanical performance (fully exposed without much change of performance), and the resultant algorithms (which does not require too much change as long as we train our model using images of the soft structure underwater) in a decoupled fashion. We believe it is easier to waterproof a camera than a force sensor based on the above reasons.
> > >
> > > We agree that vision underwater would cause some changes in the sensor reading with our design, and this is probably the beauty of using the learning method. We could introduce underwater images of the soft structure's deformation while interacting with objects underwater to develop new models and networks that learn the 6D force and torque data underwater. We are currently in the process of this work and will report further results in an upcoming paper, but it would be outside the scope of this manuscript.
> > >
> > > > I would like to clarify that when initially reading the paper, it was not clear to me if the material the finger is manufactured from is a single type of material or a composite material ... It would benefit the reader if it were clearly stated which type of material is used.
> > >
> > > We apologize for the confusion, and we will update our manuscript on the specific material used in this paper to address the reviewer's concern. We have already published multiple RAL papers on this part in our previous work using different structural designs and have filed patents on these soft structures. There are multiple variations in the structural design, and we just presented one of them in this paper for the sake of simplicity, as our focus is on soft robot force control in this paper.
> > >
> > > > Technically, the authors are correct to use the term metamaterial in their paper if they use a single type of material and vary its structure ... Other papers in the area of metamaterials used in soft robotics utilize commonly much smaller and repetitive structures, composed of individual cells or macro/micro structures.
> > >
> > > We think 3D metamaterial is the most suitable term for our soft structure design, as one can use pretty much any soft matter to build it as long as the 3D structure is presented, but would be different in its adaptive performance (this is also the core concept behind metamaterial that the structural geometry, even at a macro level, would provide properties that are not available to the original material itself). The soft structure used in this paper clearly presents a repetitive pattern of a pyramid shape from top to bottom. There are multiple variations, as presented in our previous work but not used in this paper. It should be noted that the structural geometry mainly defines a metamaterial, which can be at any scale as long as it provides a new property. The cited paper is a good example, but many metamaterial examples are published in material science at a macro scale. We did not optimize the design in this paper but only chose the most suitable one to present the concept of soft robot force control. We will revise our updated manuscript to clarify further the term of 3D metamaterial used to avoid confusion.

---

> > > > ### Comment · Reviewer_avKF · 2021-09-02
> > > > **Post-Rebuttal comment**
> > > >
> > > > After considering the detailed response by the authors, changes made to the paper and other reviews I am okay with upgrading my review to a 'weak accept'.
> > > >
> > > > In summary, the authors present in their paper an interesting application of machine learning to the problem of computing force measurements based on the visual recognition of deformations in a deformable structure with the help of deep learning.  Requested additional experiments were added, and weaknesses in readability were addressed.

---

> ### Author Response · Authors · 2021-08-27
> **Reply to Reviewer avKF (Part III)**
>
> **4. *I cannot entirely agree with the conclusion that the proposed setup would allow other researchers a more versatile or low-cost setup: Assuming reasonable variance in manufacturing or 3D-printing the proposed structure, every lab would have to carry out the calibration method described, i.e. collecting a supervised data set with force measurements and images showing the deformation. This process would also be required to be repeated every time a new finger/structure is produced*.**
>
> **[Reply 4]** We have preliminarily tested our design in various ways, which works just fine without too much trouble. In our previous works regarding the soft metamaterial's use in robotic grasping, we have demonstrated a wide range of design variations, sharing similar omni-adaptive capabilities. This paper proposes a more systematic way of calibrating the proposed structure in 6D force and torque sensing using an industrial-grade, desktop-sized robotic arm to perform the task automatically. One can even complete the task directly without involving the robot arm; just play with the soft structure while capturing the images and FT data simultaneously for later training, as we have demonstrated in this paper's setup with the tag. On the other hand, the 3D printing technologies nowadays are pretty mature to ensure a reasonable and acceptable variation in our case, which did not show much trouble in the results. We hope to establish a more robust and general-purpose baseline training for a series of standard finger designs so that other users may print them using the same method, material, and setting, then perform a few simple tasks by playing with the soft structure, ending up having an acceptable and capable 6D FT estimation ready for use. However, this would require further research and study, which we aim to achieve by leveraging what we have developed in this paper and then push for different outcomes for more general-purpose use. However, one must note that each commercial FT sensor would also need a careful calibration before shipping or even before use. Such a process is usually prepared by the manufacturer in commercial cases and not much different in ours.
>
> **5. *The paper introduced the soft structure as a 3D metamaterial. As the introduction of the paper already mentions, a meta-material is usually a composite material, i.e. a material made up of at least two or more materials. It is not clear if this is the case here, or if simply the structure of the elastic material (in the form of a pyramide like tubing) determines alone its properties*.**
>
> **[Reply 5]** We would like to clarify that meta-material is different from composite material, as cited in reference [11] in our paper. Meta-material leverages the structural design and geometric feature, which usually involves just one material (but not limited to), to achieve mechanic properties not capable by the original material. Composite ones, as explained by the reviewer, involve at least two or more materials. Our design shown in this paper is a meta-material that involves only one material, but we are also experimenting with other designs that involve multiple materials to build such meta-structure.
>
> **6. *For me, the results presented in Figure 6 (b) (UR 10 force sensor) look considerably more natural and better than (d) proposed camera-based approach*.**
>
> **[Reply 6]** We appreciate the reviewer's careful review of the details in our paper. After investigating the performance carefully, we have improved the stability of the soft sensor by utilizing a corner refinement method offered in OpenCV to detect the ArUco tag. Hence, we also did the robot teaching experiment again as shown in figure 6(a)-(d) of the updated manuscript. The performance of the soft sensor in this task has been improved greatly. Besides, we would like to point out that the experiment shown in Fig 6 is a qualitative study towards a low-cost yet reasonably capable demonstration, in which the user's interaction experience is also very important. During the experiment, the user needs to hold the whole tool flange section of the UR10e to draw the circle, which is bulky for the human hand to operate. In the contrast, it is very easy and natural to drag the soft fingertip of our design, as described by the test subject after the experiment. From both perspectives, our proposed system is superior in Fig. 6's task.

---

> ### Author Response · Authors · 2021-08-27
> **Reply to Reviewer avKF (Part II)**
>
> **2. *The estimated force/torque data is utilized to be fed back into the controller of the UR10 in task-space, controlling the position of its end-effector. Why can this not be solved with said additional force sensor at the base of the finger or a more sensitive (robot) controller? This setup would not require deep learning at all*.**
>
> **[Reply 2]** As mentioned in the reply above, one can surely use an FT sensor at the base, only more expensive, less adaptive to the changing environment, and more complex in engineering design and integration. However, what if we want to use the gripper under the water, just for the sake of argument, do we need to change/waterproof the entire design. This is only a simple example, but with our design, the problem could be solved more quickly without changing the design, which is an added benefit of our method that could work both on land and under the water or in other challenging situations.
>
> We proposed two ways to benchmark our design. One is learning-based through deep learning that directly estimates the 6D interaction force and torque through images of the soft metamaterial's internal view. The other uses a purely vision-based method with fiducial markers to calculate a particular feature of the soft structure's internal deformation during the interaction. Moreover, both methods are also calibrated with commercial 6D force and torque sensors for the sake of comparison. The apparent benefit of the learning method against the vision one is the removal of markers, which may interfere with the soft metamaterial's omni-adaptive performance.
>
> Nevertheless, detailed modeling of the metamaterial's mechanical performance is a challenging topic using analytical methods. This is where the learning-based method could contribute, especially if using it in a challenging environment. We believe our demonstration and experiment results have proved the advantage of using the learning-based method in our design, which aligns with the purpose of this conference.
>
> **3. *Why is deep learning the algorithm of choice? Why not use the information from the tag extracted with OpenCV to train an SVM or another regression algorithm? Figure 5 shows only the predictions of the used neural network, some other learning algorithm as baseline would be expected. Or an explanation of why other algorithms are not applicable*.**
>
> **[Reply 3]** Please kindly refer to our reply in the above section for the choice of algorithm. In short, we can surely use the tag for our purpose, as we have demonstrated in section 3.3. The only drawback is that the tag structure may interfere with the soft metamaterial's omni-adaptive capability and limit the application scenarios, which can be addressed when using the learning-based method through images of the whole finger's internal deformation. We also used the tag, or the vision-based method, to compare to the learning-based one and found comparable performance between them. One can surely pick other neural networks to improve the results or even better FT sensors for an improved benchmark. We would like to further address this issue in a forthcoming paper, but keep our CoRL submission with a clean and direct demonstration to show the feasibility of our design. This work needs further development in the choice of learning-based algorithms and use integration, which we are developing as the next step. We think other algorithms could also work, but as a baseline of research, a common one such as resnet should demonstrate its primary performance and engineering potential, which we are trying to achieve in this paper.

---

> ### Author Response · Authors · 2021-08-27
> **Reply to Reviewer avKF (Part I)**
>
> We sincerely thank the reviewer's constructive comments on our submission. Moreover, we appreciate the reviewer's positive recognition of our proposed method and demonstration. We agree with the reviewer that using a reinforcement learning setup would potentially provide more interesting results as a learning problem. In this paper, we felt that it would be more fundamental to address the feasibility of directly learning a soft structure's interaction force and torque through vision sensors and achieving comparable performances against commercial products commonly used in force-sensitive robotic manipulations. We have filed a patent for our design and are further integrating advanced learning methods, including unsupervised ones, to further explore this subject.
>
> From a soft robotic perspective, we have seen many designs focusing on tactile sensory with vision integration through a layer of soft material. However, a tight integration within the geometric constraints on the gripper is still tricky. Our method conveniently integrates the whole finger body and 6D force-torque sensing through vision and learning, which is not published in the current literature. In most industrial pneumatic or electric grippers, the whole finger section is usually not provided, and customized finger design is usually involved, so simply adding a layer of sensitive skin may not be enough. Using our method, one can quickly introduce omni-directional adaptation to these rigid grippers with added sensory in 6D force and torque within a compact form and low cost. From this perspective, we believe that our design has contributed to integrating the learning-based method for robotic research, even if the presented method is supervised.
>
> Below are replies to address comments raised by the reviewer.
>
> **1. *The general learning problem is not very well formalized: What exactly is the learning problem the paper tries to tackle? Just as an example, the paper does not mention that the learning setup is supervised at least once. The greater question is, why is the learning problem supervised? Why not unsupervised or using reinforcement learning? Furthermore, the ATI Nano25 sensor appears smaller than the camera box at the bottom of the finger. Why not simply attach the ATINano25 sensor instead of a camera? What extra information would the camera provide, and why would we be unable to use an F/T sensor at the base*?**
>
> **[Reply 1]** We apologize for neglecting a formal explanation of the learning problem we were trying to address in this paper. While many previous works focused on learning tactile feedback through a layer of soft skin, we were trying to tackle the feasibility of direct learning the whole finger adaptation through learning during the interaction, namely the learning-based proprioception of soft robot interaction. Such a proprioceptive sensing capability is still a challenging subject in soft robotic research, which we have presented a very feasible solution with proven capability. The learning problem is to estimate the force and torque at the base of the soft sensor from the deformation of the metamaterial, which is a regression problem and can be solved nicely with the supervised learning method [1]. We have added the problem definition in section 3.2.
>
> Sensors from suppliers such as ATI seem smaller in design but significantly more expensive. They have a limited use scenario if we want to put our finger/gripper in an extreme environment, such as underwater grasping. The camera box used in this paper is an early engineering prototype, which can be significantly smaller in size and dimension through different designs. We are currently working on a much more compact design with a more miniature camera, which is outside the scope of this paper.
>
> All we need from the camera is the image of the whole-body deformation of the soft metamaterial from an unobstructed view angle to use a learning-based method to estimate the 6D interaction force and torque in real-time. We believe this is highly relevant to the purpose of this conference. One can surely use an FT sensor at the base, only more expensive, less adaptive to the changing environment, and more complex in engineering design and integration.

---

### Official Review · Reviewer_Wc5U · 2021-07-23

**Originality:** Good
**Technical Quality:** Very Good
**Clarity Of Presentation:** Very Good
**Impact:** 4

**Recommendation:**

Weak Accept: I recommend accepting the paper, but will not argue for my recommendation if the majority of other reviewers have a different opinion.

**Summary:**

The paper explored the use of soft material for robot force sensing and control. The main contribution is the introduction and analysis of a cone-shaped metamaterial, and two vision-based approaches to infer the applied force on the material based on deformation. The proposed system is evaluated on a soft robot force control task.


**Issues:**

Please see weaknesses.


**Reviewer Expertise:**

Good: General knowledge of the area

**Strengths And Weaknesses:**

Strengths:
1. The idea of leveraging soft materials and computer vision for robot force control is interesting and reasonable.
2. The proposed metamaterial structure is concrete and carefully analyzed.
3. Two different methods were evaluated to learn the force profile of the soft material under different forces and are both shown to be effective, though one has some generalization issue.
4. Preliminary real-world experiments were conducted to demonstrate the usefulness of the proposed idea.

Weaknesses:
1. Limitations of the proposed method are not well discussed. For example, what’s the highest frequency that the proposed system can run at? A force-torque sensor can usually run at very high frequency but a camera may have limited frequency. Another potential limitation is that the system would be more sensitive to lighting conditions and also introduce a higher cost of computation in exchange for lower cost in hardware. Some discussion along these lines would be helpful.
2. The proposed system didn’t demonstrate superior task performance in the evaluated task: in Figure 6, it seems that with force-torque sensors the resulting trajectory is smoother.
3. For the AR-tag approach, is it able to infer pulling or push forces on the material? For example if the robot hits a wall vertically, it would be important to properly detect that.


**Summary Of Recommendation:**

I think the idea presented in this paper is interesting and promising in my knowledge. The design of the meta-material and the presented learning approach to infer the force and torque readings are solid. On the other hand, some more discussions are needed and a better experiment setup would make the paper even stronger and more interesting. For example, if one can build a two-finger gripper out of the proposed method, it’ll be able to sense forces per-finger. This is something difficult to achieve with existing methods and can potentially improve manipulation performance, which would better support the value of the paper.

---

> ### Author Response · Authors · 2021-08-26
> **Reply to Reviewer Wc5U**
>
> We sincerely thank the reviewer's constructive comments to our submission. Furthermore, we appreciate the reviewer's positive recognition of the strengths and contributions of this paper, including its limitations.
>
> Below are our replies to the weaknesses raised by the reviewer.
>
> **1. *Limitations of the proposed method are not well discussed. For example, what’s the highest frequency that the proposed system can run at? A force-torque sensor can usually run at very high frequency but a camera may have limited frequency. Another potential limitation is that the system would be more sensitive to lighting conditions and also introduce a higher cost of computation in exchange for lower cost in hardware. Some discussion along these lines would be helpful*.**
>
> **[Reply 1]** We agree with the reviewer for our lack of discussion on the limitations of our proposed method. We have updated section 4, paragraphs 2 and 3 in our updated manuscript to reflect the reviewer's concerns. We agree with the reviewer that the camera's frame rate, system cost, and environment lighting condition could play a significant role in our design's performance. However, our result shows that our system is reasonably robust against these factors, as shown in the added content.
>
> **2. *The proposed system didn’t demonstrate superior task performance in the evaluated task: in Figure 6, it seems that with force-torque sensors the resulting trajectory is smoother*.**
>
> **[Reply 2]** We thank the reviewer for mentioning the manipulation tasks. After investigating the performance carefully, we have improved the stability of the soft sensor by utilizing a corner refinement method offered in OPENCV to detect the ArUco tag. We also conducted the robot teaching experiment again, as shown in Figure 6(a)-(d) in the updated manuscript, showing a significant improvement in the soft sensor's task performance.
>
> Besides, we would like to point out that the experiment shown in Fig 6 is a qualitative study towards a low-cost yet reasonably capable demonstration, in which the user's interaction experience is also very important. During the experiment, the user needs to hold the whole tool flange section of the UR10e to draw the circle, which is bulky for the human hand to operate. In the contrast, it is very easy and natural to drag the soft fingertip of our design, as described by the test subject after the experiment. From both perspectives, our proposed system is superior in Fig. 6's task.
>
> **3. *For the AR-tag approach, is it able to infer pulling or push forces on the material? For example, if the robot hits a wall vertically, it would be important to properly detect that*.**
>
> **[Reply 3]** We are glad that the reviewer noticed this detail. With the AR-tag approach, one can infer pulling or pushing forces to some extent with our design, but not at its best if using more rigid material when fabricating the metamaterial. The metamaterial design used in this paper does not have the best adaptation along its line of symmetry along its longitudinal direction. One can either use softer material or use other metamaterial designs with a more extensive deformation along this direction to achieve an improved force estimation when pulling or pushing. We successfully tested it using other metamaterial designs and will discuss this further in a forthcoming paper.  However, this should not affect the essence of our proposed method. For the 3D metamaterial used in this paper, one could detect hitting-the-wall-vertically by detecting sudden force change.
>
> **4. *I think the idea presented in this paper is interesting and promising in my knowledge. The design of the meta-material and the presented learning approach to infer the force and torque readings are solid. On the other hand, some more discussions are needed and a better experiment setup would make the paper even stronger and more interesting. For example, if one can build a two-finger gripper out of the proposed method, it’ll be able to sense forces per-finger. This is something difficult to achieve with existing methods and can potentially improve manipulation performance, which would better support the value of the paper*.**
>
> **[Reply 4]** We sincerely appreciate the reviewer's recognition of our work. We agree that adding a two-finger gripper experiment would be an interesting demonstration, but difficult for the limited time. However, we felt that our current experiment already demonstrated a comparable and capable performance of our proposed design in robot force control against the commercial products used in this paper and many research scenarios. The tasks achieved using our design is (probably) the most common demonstration with a commercial 6D force-torque sensor installed at the robot tool flange (the e-series of UR10 is a perfect example, where a 6D FT sensor is commercially integrated inside its tool flange as a standard design), which is well-demonstrated in our paper and the submitted video.

---

> > ### Comment · Reviewer_Wc5U · 2021-09-02
> > **Post rebuttal**
> >
> > The detailed response from author is very much appreciated. After reading the rebuttal and other reviews, I will keep my current position of weak accept.

---

### Official Review · Reviewer_h8MY · 2021-07-25

**Originality:** Very Good
**Technical Quality:** Very Good
**Clarity Of Presentation:** Very Good
**Impact:** 4

**Recommendation:**

Strong Accept: I recommend accepting the paper and will argue for my recommendation even if other reviewers hold a different opinion.

**Summary:**

The paper develops a soft finger and leverages learning to estimate force/displacement from visual observation of deformations during interaction with external objects.

**Issues:**

Issues are listed in recommendation.

**Reviewer Expertise:**

Excellent: Expert knowledge on the topic of the paper

**Strengths And Weaknesses:**

Strengths:
1. The paper is leveraging learning to enable force estimation from deformations of a 3d printed finger leveraging a camera. This could enable low cost fingertips for use in the industry where the fingertips can be swapped out with minimal cost.
2. Designing these fingertips to have 6d force/torque estimation is very interesting and leveraging learning to go around modeling the deformations has great benefits as shown by the paper.
3. The teleop application shown with the proposed method is very cool and shows a direct application of the approach.


Weakness:
1. The paper learns a mapping from finger deformation to 6D force/torque. However the accuracy of the predictions is not validated. Plotting the error in predictions both in magnitude and direction would be helpful. Previous force estimation methods have reported errors on test dataset to help compare across methods [biotac_force].
2. Does the learned model transfer to a new sensor of the same design? Discussing this question with some experiments is needed.
3. The paper does not cite any existing work on tactile force estimation. Specifically, the idea of collecting data leveraging robots and learning a NN is not novel, it has been used to estimate force and deformation fields in previous works[biotac_force, biotac_physics].
4. Why are the predictions very noisy? Is there any filtering that is done post prediction to enable smooth teleop?
5. The paper doesn’t show what happens when force is applied in non-normal directions, i.e. between parallel (shear) and normal angles. How does the accuracy change when force is applied in these directions? While estimating shear force is not necessary, having accurate force angle prediction is important for many manipulation tasks[biotac_directionality].  Showing some plots when force is applied in these directions is required to better understand the limitations of the work.
6. Can the force estimated be used for manipulation tasks such as gentle object placement? [biotac_force]


References:

[biotac_force] Sundaralingam, Balakumar, Alexander Sasha Lambert, Ankur Handa, Byron Boots, Tucker Hermans, Stan Birchfield, Nathan Ratliff, and Dieter Fox. "Robust learning of tactile force estimation through robot interaction." In 2019 International Conference on Robotics and Automation (ICRA), pp. 9035-9042. IEEE, 2019.

[biotac_physics] Narang, Yashraj S., Karl Van Wyk, Arsalan Mousavian, and Dieter Fox. "Interpreting and Predicting Tactile Signals via a Physics-Based and Data-Driven Framework."

[biotac_directionality] Gutierrez K, Santos VJ. Perception of Tactile Directionality via Artificial Fingerpad Deformation and Convolutional Neural Networks. IEEE Trans Haptics. 2020 Oct-Dec;13(4):831-839. doi: 10.1109/TOH.2020.2975555. Epub 2020 Dec 25. PMID: 32092014.


**Summary Of Recommendation:**

The paper does not introduce any novel learning/data collection methods. But leverages learning to enable force estimation from hard to model deformations of a fingertip. There could be several applications of the proposed solution. Addressing the following is required to improve the recommendation:
1. Reporting magnitude and directional accuracy.
2. Analyzing transfer across other fingertips of the same design.
3. Analyzing accuracy of force estimation in non-normal directions.
4. Showing a force controlled task such as grasped object placement based on detection of object contact with table.

---

> ### Author Response · Authors · 2021-08-26
> **Reply to Reviewer h8MY (Part II)**
>
> **[Issue 3] "*Analyzing accuracy of force estimation in non-normal directions*."**
>
> **[Reply 3]** We agree that analysis in non-normal directions could be an essential aspect of our proposed design. However, as explained in our reply to the first issue, our proposed method is in direct comparison to classical force-torque sensors, which are usually installed on the tool flange whereas the tactile sensors are often installed on the finger surface. Therefore, results on this part may not serve a useful purpose in our case. Nevertheless, as shown in figure 6(e), the force applied by the human hand is in non-normal directions, and the plots in figure 6(e) show the force along all axes at the base of the finger structure. In the human-robot interaction experiment using UR10e, the soft structure can deform and sense the 6D force and torque signal at the finger base. As pointed in the final remarks, we will investigate its deformations more systematically to include further analysis on its force estimation at the surface of interaction, in a similar setup as most tactile sensors, in our future work, which is outside the scope of this paper.
>
> **[Issue 4] "*Showing a force controlled task such as grasped object placement based on detection of object contact with table*."**
>
> **[Reply 4]** We thank the reviewer for pointing out tasks such as object manipulation. As explained in our reply to Issue 1, our proposed method mainly deals with a vision-based method with learning models that could potentially replace conventional force-torque sensors at the robot flange using a soft metamaterial structure. We systematically demonstrated such comparison in our experiment in Fig. 6.
>
> Contact detection during object grasping is an exciting task that could leverage force control capabilities for advanced skills. However, our paper is mainly to validate its force-torque sensing capability against commercial FT sensors on the robot tool flange. We intend to add the suggested experiment in our future work that emphasizes grasp detection and dexterous manipulation with "soft robot force control," as proposed in this paper.
>
> We felt that our current experiment already demonstrated a comparable and capable performance of our proposed design in robot force control against the commercial products used in this paper and many research scenarios. The tasks achieved using our design is (probably) the most common demonstration with a commercial 6D force-torque sensor installed at the robot tool flange (the e-series of UR10 is a perfect example, where a 6D FT sensor is commercially integrated inside its tool flange as a standard design), which is well-demonstrated in our paper and the submitted video.
>
> **For the weaknesses raised by the reviewer.**
> 1. Please refer to our reply to Issue 1 as above.
> 2. Please refer to our reply to Issue 2 as above.
> 3. Please refer to the beginning section of our reply, which is all cited.
> 4. In the manuscript, the prediction plotted in figure 6(e) is the raw data without any filter to reflect the raw performance of the sensor. The noises mainly came from visual detection of the tag. In the updated manuscript, we have improved the stability of the tag detection by using a corner refinement method offered in OPENCV. As reported in section 3.3 paragraph 4, the standard deviations of the predicted signals (Fx, Fy, Fz, Mx, My, Mz) of the soft sensor measured over 100 seconds are (0.009, 0.034, 0.007) N and (0.0013, 0.0009, 0.0010) Nm. The same measurement for UR10e are (0.323, 0.431, 0.253) N and (0.0123, 0.0107, 0.0058) Nm. The noise level of the soft sensor is about 10 times smaller than that of the UR10e. In the teleoperation application, as demonstrated in the paper, we took a simple moving average of the tag's displacement to make the control smoother. We also conducted the robot teaching experiment again as shown in Figure 6(a)-(d). The performance of the soft sensor in this task has been improved.
> 5. Please refer to our reply to Issue 3 as above.
> 6. Please refer to our reply to Issue 4 as above.

---

> ### Author Response · Authors · 2021-08-26
> **Reviewer h8MY (Part I)**
>
> We sincerely thank the reviewer's constructive comments on our submission. We appreciate the reviewer's positive recognition of the strengths and contributions of this paper. We also agree with the reviewer's suggestion and added the listed references to our paper. While the method proposed in this paper has roots in previous work, we still believe that our paper explores a novel adoption of the learning method in robot force control through the medium of soft material and structures. We believe that our paper holds the potential to open new applications and sensory integration in the robotic system for a broader range of interactions with the human operators through the concept of learning-based force control using soft robot design.
>
> The novelty of our design mainly lies within our proposed use of vision-based learning through a soft metamaterial deformation during physical interaction, making the overall design more robust in some particular and challenging scenarios. For example, for grippers to be operated underwater, classical force-torque sensors would require a complex waterproofing design with increased cost and complexity, which is similar for most tactile sensors such as BioTac. However, our proposed method may perform well without change in design, as long as we waterproof the camera case, which can be shared between on-land and underwater uses (without any modification in the mechanical parts). However, we did not emphasize this part in our current paper for the sake of the scope, which we will address with more details in another forthcoming paper to be published elsewhere soon. Nevertheless, the concept should be straightforward to understand without interfering with the aim of this paper.
>
> Below are our replies to the issues raised by the reviewer.
>
> **[Issue 1] "*Reporting magnitude and directional accuracy*."**
>
> **[Reply 1]** We agree with the reviewer that it is essential to investigate both magnitude and directional accuracy. However, it is also important to clarify that our proposed method mainly deals with the force and torque estimation at the finger's base instead of at the contact interface. This is a critical difference from other tactile-centric sensors, with or without visions. As shown in the last experiment in our paper, we compare our result directly to a commercial force-torque sensor commonly installed at the tool flange. This is a more reasonable comparison in our case, and due to the use of the soft structure as the main interface of interaction, we explore the concept of "soft robot force control" in direct comparison with classical force-torque sensors performance at the tool flange. Our result shows a qualitatively comparable performance using our proposed method against commercial FT sensors. However, due to the limited performance of the force-torque sensor on UR10e, the prediction discrepancy between the soft sensor and UR10e is not an exact quantity to evaluate the performance. During the experiment, we have observed zero drift on UR10e quite often. To partly address the reviewer's concern, we compared the signal noise measurement of our proposed design to that of UR10e in the updated manuscript in section 3.3 paragraph 4, which is a standard measurement commercial F/T sensor would provide.
>
> **[Issue 2] "*Analyzing transfer across other fingertips of the same design*."**
>
> **[Reply 2]** We agree with the reviewer that it is crucial to validate if the learned model is transferred to the new sensor of the same design. Thus, we have fabricated two fingers of the same design, material, and fabrication methods. One was used to collect the training data and the other was used to test the performance. We conducted a new experiment as described in section 3.3 paragraph 4 of the updated manuscript and compared the prediction of the test finger and F/T readings from UR10e in figure 6(e). The results show that the learned model can be transferred to the new sensor of the same design very well.

---

> > ### Comment · Reviewer_h8MY · 2021-09-04
> > **Thanks for clarification [Score upgrade]**
> >
> > Thanks you for the clarifications. This paper is leveraging learning to overcome hard to model deformations and I think this will have a profound impact in how tactile perception will be used in robotics. I have upgraded my score as I believe the rebuttal has addressed most of the concerns I had.
> >
> > The only concern I have is that it would still be good to report the mean and median error (across independent dims) between the predictions and the ground truth from the test set of the collected dataset with the ATI sensor. I think r^2 in fig 6 already addresses this but stating in mean, median is still useful as most tactile sensors and F/T sensors report accuracy along the individual axis. Some tactile sensors have shown to not have good magnitude accuracy but still scale well (can differentiate between low and high force) and hence it would good to know if the proposed sensor also has this bottleneck.

---

### Meta-Review · Area_Chair_k1pp · 2021-08-14

**Recommendation:** Accept (Poster)
**Confidence:** 5

**Metareview:**

The paper proposes achieving soft robot force control using a 3D flexible structure and learning estimates of force-torques by using vision, potentially decreasing the cost of robotic systems. In general, the paper has been considered of having a good impact, originality, and technical quality.

Amongst others, reviewers have raised issues regarding deficiencies in the related work, confusion concerning effectiveness in the estimation of forces in certain directions (vertically hitting a wall), and whether the performance against the conventional force-torque sensor is truly comparable. Additionally, it seems that a bit more motivation and more details for the CNN have been missed. The authors are encouraged to discuss the raised points with the reviewers during the rebuttal.

Post rebuttal ==============

The paper has been gone through extensive discussions between authors and reviewers. The overall scores have been increased during the rebuttal as the authors clarified most of the concerns. As a final result, we observe an improved final manuscript.

---

> ### Author Response · Authors · 2021-08-26
> **Reply to Area Chair k1pp**
>
> We sincerely appreciate the constructive comments provided by all reviewers, and we are glad to see the diverse and in-depth discussion from the reviewers and AC on our work from various perspectives. We agree with the reviewers that there is still a lot to improve in our manuscript, which we have revised extensively based on the comments. From the reviewers' comments, we truly believe that our paper aligns nicely with the conference's purpose, and we hope to be published at CoRL to generate more discussion on our work, just as what we have seen from the reviewers' comments. We believe that soft robot force control could be an exciting area of research in robot learning that combines soft robot design through meta-material and machine vision techniques with learning-based methods. This subject is yet to be explored in the literature, which we hope to publish through the platform that CoRL has created to share with a more relevant community in robot learning. Please refer to the detailed reply below, and we sincerely wish a favorable decision from the AC and reviewers on our submission. Thanks~

---

### Decision · Program_Chairs · 2021-09-13

**Decision:**

Accept (Poster)

**Comment:**

The paper proposes achieving soft robot force control using a 3D flexible structure and learning estimates of force-torques by using vision, potentially decreasing the cost of robotic systems. In general, the paper has been considered of having a good impact, originality, and technical quality.

Amongst others, reviewers have raised issues regarding deficiencies in the related work, confusion concerning effectiveness in the estimation of forces in certain directions (vertically hitting a wall), and whether the performance against the conventional force-torque sensor is truly comparable. Additionally, it seems that a bit more motivation and more details for the CNN have been missed. The authors are encouraged to discuss the raised points with the reviewers during the rebuttal.

Post rebuttal ==============

The paper has been gone through extensive discussions between authors and reviewers. The overall scores have been increased during the rebuttal as the authors clarified most of the concerns. As a final result, we observe an improved final manuscript.